# The Application of Regenerated Silk Fibroin in Tissue Repair

**DOI:** 10.3390/ma17163924

**Published:** 2024-08-07

**Authors:** Zhaoyi Li, Guohongfang Tan, Huilin Xie, Shenzhou Lu

**Affiliations:** National Engineering Laboratory for Modern Silk, College of Textile and Clothing Engineering, Soochow University, Suzhou 215123, China; 20224215024@stu.suda.edu.cn (Z.L.); 20234015007@stu.suda.edu.cn (G.T.); 20225215095@stu.suda.edu.cn (H.X.)

**Keywords:** silk fibroin, soft materials, wound repair, tissue regeneration, hydrogels, porous scaffolds

## Abstract

Silk fibroin (SF) extracted from silk is non-toxic and has excellent biocompatibility and biodegradability, making it an excellent biomedical material. SF-based soft materials, including porous scaffolds and hydrogels, play an important role in accurately delivering drugs to wounds, creating microenvironments for the adhesion and proliferation of support cells, and in tissue remodeling, repair, and wound healing. This article focuses on the study of SF protein-based soft materials, summarizing their preparation methods and basic applications, as well as their regenerative effects, such as drug delivery carriers in various aspects of tissue engineering such as bone, blood vessels, nerves, and skin in recent years, as well as their promoting effects on wound healing and repair processes. The authors expect SF soft materials to play an important role in the field of tissue repair.

## 1. Introduction

### 1.1. Importance of Wound Repair

The skin, as the largest organ by surface area in the human body, helps protect the body from various harmful factors in the external environment, such as physical, chemical, mechanical, or pathogens, acting as an external barrier to maintain and coordinate the stability of the internal environment. Therefore, the intact structure and function of the skin are of great significance for maintaining normal physiological activities [1,2]. However, people inevitably suffer from skin damage and dysfunction due to burns, mechanical trauma, and chronic diseases (such as diabetic ulcers). This leads to weakened or lost normal physiological functions of the skin, exposing the body directly to various pathogenic environments, and greatly increasing susceptibility to diseases [3]. Skin defects ultimately lead to metabolic disorders, tissue necrosis, and other adverse consequences, and, in severe cases, can be life-threatening. Chronic wounds are typically difficult to treat, have long treatment cycles, and are expensive, causing a huge economic burden on patients’ families and society. Approximately 1% of the world’s population suffers from trauma, and about 5% of medical expenses are spent on wound repair. By 2024, the global wound care market is expected to have a compound annual growth rate of 4.6%, increasing from 19.8 billion USD in 2019 to 24.8 billion USD in 2024 [4].

### 1.2. The Role of Soft Materials in Wound Repair

For skin damage caused by diseases or accidents, traditional treatment methods mainly include autologous transplantation, allogeneic transplantation, and xenotransplantation. However, these are limited by the availability of donors and the potential for immune rejection [5]. Tissue engineering technology aims to design, construct, improve, and cultivate bioactive implants for defective tissue repair, providing a possibility for tissue transplantation. Seed cells, scaffolds, and growth factors are the three main elements of tissue engineering. Among them, tissue engineering materials not only replace the extracellular matrix, providing a substrate for seed cell adhesion and a place for metabolite and nutrient exchange, but also provide stimuli for cell adhesion, migration, proliferation, and differentiation, and regulate the structure of generated tissues [6]. Suitable tissue engineering materials have spatial structures and mechanical properties that are compatible with repair tissues and newly formed tissues, and their degradation products are non-toxic and can be absorbed by the body or safely eliminated. Additionally, they have good biocompatibility and do not cause significant immunogenic reactions when implanted in the body [7].

### 1.3. Current Soft Materials Used in Tissue Repair

Natural polymers used in tissue repair mainly include collagen, gelatin, alginate, chitosan, and SF. Collagen is the most widely distributed functional protein in mammals, with a unique multi-level self-assembly structure. It has suitable mechanical properties, diverse biological functions, and enzymatic degradation properties [8]. However, collagen faces several challenges in tissue repair applications, such as low yield, poor thermodynamic stability, and rapid degradation by collagenase in the body. Additionally, the synthesis cost of collagen is high, making large-scale production difficult. Methacrylate gelatin (GelMA) hydrogels are often used in wound repair and cartilage tissue repair. GelMA combines the characteristics of natural and synthetic biomaterials, providing a three-dimensional structure suitable for cell growth and differentiation, excellent biocompatibility, and can replace artificial basement membranes or other natural collagen hydrogels [9]. Despite its potential in tissue repair, GelMA faces challenges like relatively weak mechanical properties and stability, which may limit its widespread application in cartilage tissue. Though it has good biocompatibility and biodegradability, long-term application may trigger immune and inflammatory responses. Alginate is the most abundant marine biopolymer, second only to cellulose. It includes potassium alginate, magnesium alginate, sodium alginate, and their corresponding ammonium and calcium salts. In recent years, there have been numerous reports on the application of alginate in tissue engineering materials [10]. Studies have shown that alginate has excellent biocompatibility, good biodegradability, and no immunogenicity. However, it still has drawbacks such as weak mechanical strength, a lack of cell-specific binding sites, and the ease with which calcium salt scaffold structures are destroyed in physiological environments. Chitosan, a product of chitin deacetylation, has advantages such as low toxicity and good degradability. Research has found that chitosan can induce the release of substances like platelet-derived growth factor and β-thromboglobulin, thereby promoting platelet activation and aggregation, effectively aiding in wound hemostasis. Additionally, chitosan can inhibit various pathogens and assist in granulation tissue formation, thus accelerating wound healing. However, chitosan-based biomaterials have low mechanical strength, usually requiring other polymers to enhance their mechanical properties [11].

### 1.4. Advantages of SF Soft Materials in Tissue Repair

SF protein, as a natural fibrous protein from silkworms, can be degraded both in vitro and in vivo, with degradation products being amino acids or oligopeptides, which are easily absorbed by the body without toxic side effects on tissue cells [12]. SF has good biocompatibility, excellent mechanical properties, and low immunogenicity [13]. SF materials are easily moldable and can be made into various forms such as films, gels, and sponges to adapt to different types of tissues. Additionally, SF is a Food and Drug Administration (FDA)-approved material currently used in many cosmetic and medical applications. Due to its excellent biocompatibility and bioactivity, SF soft materials have broad application prospects in wound repair.

## 2. Stages Involved in the Wound Repair Process

Wound repair is a complex process involving interactions with different cells and matrices, as well as various overlapping stages, including hemostasis, inflammation, new tissue formation, and tissue remodeling. The process can be roughly divided into four stages in chronological order: hemostasis, inflammation, proliferation (new tissue formation), and maturation (tissue remodeling) (Figure 1) [14]. As one of the most complex biological processes in the human body, wound repair involves the regulation of a series of complex cellular behaviors and the control of the wound microenvironment. The inflammatory phase following hemostasis is the first stage of wound healing, occurring immediately after injury and lasting up to 2 days. It requires activation of inflammatory pathways, coagulation cascades, and the immune system to prevent continuous loss of blood and bodily fluids. Inflammatory cells such as neutrophils and macrophages remove pathogens or damaged cells through phagocytosis and produce various cytokines and growth factors [15]. New tissue formation is the next stage of wound healing, associated with angiogenesis, re-epithelialization, granulation tissue configuration, matrix/collagen deposition, and wound contraction [16]. Depending on the extent of the injury, the tissue remodeling phase can last for a year or longer and is combined with the remodeling of the extracellular matrix (ECM) [17]. Therefore, active soft materials can regulate cell behaviors, control the wound microenvironment, and have broad application prospects in accelerating wound healing.

The wound microenvironment can be broadly defined as the external environment directly adjacent to the wound surface and the internal region adjacent to the area below the wound surface. As shown in Figure 2, the antioxidant doped hydrogel can effectively reduce reactive oxygen species (ROS) mediated cell death, inhibit the proliferation of skin-related cells (such as keratinocytes, fibroblasts, and endothelial cells), and induce M1 macrophages to polarize into M2 phenotype, alleviate excessive inflammation and promote proliferation, epithelization, collagen deposition, angiogenesis, and diabetes wound healing [18]. Post-injury tissue continuously produces various cytokines, which play a role in wound repair [19].

### 2.1. Hemostasis Phase

Uncontrolled bleeding following trauma is a major cause of death [21]. Within the first few minutes of injury, damaged blood vessels rapidly constrict, and platelet receptors interact with ECM proteins (such as collagen, fibronectin, etc.), leading to platelet activation and aggregation [22]. The key to hemostasis is the adhesion, activation, and aggregation of platelets. During primary hemostasis, platelets and a small number of blood cells adhere to the internal subcutaneous collagen. This adhesion quickly activates other platelets in the blood, triggering irreversible aggregation [23]. Activated platelets combine with various clotting factors to promote the production of thrombin. With the help of transglutaminase (FX III), thrombin catalyzes the transformation of fibrinogen into fibrin, promoting the coagulation process [24]. Additionally, calcium ions aid in blood coagulation during the hemostasis stage by promoting the formation of a platelet plug. Calcium ions can trigger the intrinsic coagulation cascade reaction along with other clotting factors, thereby accelerating the synthesis of sufficient thrombin and promoting early fibrin formation [25]. Calcium ions mediate the binding of tenase and prothrombinase, which are essential for the stable incorporation of platelets into the developing thrombus [26]. The complex hemostasis cascade provides numerous targets for developing bioactive materials. These materials can promote platelet activation and aggregation, thereby facilitating wound closure. Furthermore, surface charge can also be an important factor in regulating platelet behavior. This is because, when blood cells come into contact with a positively charged surface, they rapidly aggregate. Activated platelets have relatively large sizes and surface areas and undergo morphological changes. In this context, blood cells can quickly form clots under the action of fibrin [27]. Therefore, using the surface charge of bioactive materials to directly aggregate blood cells for participation in the coagulation system may be a feasible hemostatic method. Varshney et al. [28] prepared SF/soy protein isolate blend membranes using electrospinning technology. Experiments on full-thickness skin wounds in rats showed that this blend membrane had high hemostatic properties, exhibited excellent rapid hemostatic performance, and was conducive to wound healing. Cheng et al. [29] prepared high-absorbency dry hydrogels using photo-crosslinking SF. By rapidly absorbing water, they concentrated the blood at the wound bleeding site, promoting platelet aggregation, triggering the hemostasis cascade reaction, and achieving rapid hemostasis.

### 2.2. Inflammatory Phase

Skin damage mainly activates easily stimulated non-transcription-dependent pathways, including purified molecules, ROS gradients, and calcium ion channels. Additionally, injured cells can secrete damage-associated molecules (DAMPs, such as polypeptides, adenosine triphosphate, deoxyribonucleic acid (DNA), uric acid, and ECM components), lipid mediators, hydrogen peroxide (H_2_O_2_), and chemokines, which also help recruit inflammatory cells, especially neutrophils [30]. Neutrophils are not common in normal skin, and functionalized neutrophils can activate macrophage excretion through the Ras-related C3 botulinum toxin substrate 1 (Rac1)—dependent pathway via Cysteine-rich protein (CCN1, original name CYR61, a secreted extracellular matrix (ECM)—associated molecule, belonging to the matricular CCN family). Interestingly, the complete elimination of neutrophils through expulsion marks the beginning of inflammation resolution. However, neutrophils do not appear to play a dominant role in skin wound healing. Instead, their persistence can lead to prolonged inflammation. Therefore, it is crucial to timely remove them. Macrophages can phagocytize apoptotic neutrophils. The macrophages that play a role in wound healing mainly differentiate from monocytes in the wound. These pro-inflammatory and bactericidal macrophages in the early stages of wound healing are of the M1 phenotype. They secrete pro-inflammatory factors, phagocytize pathogens, and digest ECM and thrombi [31]. As inflammation subsides, the pro-inflammatory phenotype of M1 macrophages transitions to an anti-inflammatory phenotype, the M2 phenotype. Promoting the transition of macrophages to the M2 phenotype in the later stages of wound healing can enhance tissue regeneration efficiency.

### 2.3. Proliferative Phase

The tissue regeneration and collagen deposition stage involves various events, such as the formation of the epidermal layer (re-epithelialization), construction of blood vessels (angiogenesis), and temporary formation of ECM (granulation tissue deposition). The epidermis is mainly composed of keratinocytes and is continuously renewed through the proliferation and differentiation of stem cells. About 2–3 days after damage, interfollicular epidermal stem cells from the wound edge, nearby sebaceous glands, or hair follicle bulge begin to proliferate, producing enough cells to fill the wound site [32]. When using biomaterials, keratinocytes migrate from grafts or wound edges to bridge the open wound, achieving complete epithelial regeneration. Therefore, it is necessary to create a microenvironment that allows keratinocytes to be effectively recruited and migrated. Hyaluronic acid (HA) can regulate keratinocyte activity, triggering specific signaling pathways that lead to migration and proliferation. Xuan et al. [33] prepared a coating for tissue repair using a layer-by-layer self-assembly method. The coating contained β-cyclodextrin-modified SF and adamantane-modified HA. This coating exhibited excellent antibacterial properties and biocompatibility, and the cell proliferation rate was enhanced. Angiogenesis begins with the stimulation of endothelial cells (ECs) through pro-angiogenic factors [34]. During angiogenesis, these cells can become tip cells at the forefront or stalk cells at the rear, with their development direction manipulated by Notch signaling, mainly regulated by vascular endothelial growth factor (VEGF). The formation of new blood vessels is crucial for effective wound healing, as it is necessary for nutrient delivery, oxygen homeostasis maintenance, cell proliferation, and tissue regeneration [18]. ROS are present around the wound. When ROS concentration is high, they are usually cytotoxic and can cause EC dysfunction, chronic inflammation, and oxidative stress. However, they can influence cell proliferation, migration, and differentiation at low concentrations. In the stage of wound healing, new connective tissue and granulation tissue are formed simultaneously, including tissue re-epithelialization, neovascularization, and immune regulation processes. Granulation tissue is mainly composed of fibroblasts, which can synthesize new ECM and assist in wound contraction. The granulation tissue contains newly synthesized ECM, newly formed blood vessels, and some inflammatory cells. During wound remodeling, granulation tissue is eventually replaced by normal connective tissue [35].

### 2.4. Maturation Phase

Regulating the matrix remodeling stage of wounds using biomaterials is roughly based on two ideas: promoting wound contraction and matrix deposition in the early stages and mitigating the inflammatory response in the later stages, limiting the excessive proliferation of myofibroblasts and bone marrow cells to reduce scar formation. The remodeling phase of wound healing occurs at the end of the proliferative phase. During this period, keratinocytes participate in wound re-epithelialization, while fibroblasts and ECs are responsible for ECM deposition [36]. Regulating the behavior of fibroblasts and myofibroblasts through biomaterials greatly promotes wound contraction and closure in the early stages of matrix remodeling. Li et al. [37] prepared an insulin-loaded SF porous scaffold material for treating wounds. Studies showed that two weeks later, using the insulin-loaded SF porous scaffold significantly accelerated rat wound healing and improved the wound healing rate.

## 3. Structure and Performance of SF Soft Materials

### 3.1. Structure of SF

Silk contains two main proteins: SF and sericin. Sericin is a water-soluble protein located on the outer part of silk. The interior of silk consists of two parallel-arranged, triangular SF fibers. SF is a structural protein mainly composed of two types of polypeptide chains: the heavy chain (H-chain, molecular weight 350 kDa) and the light chain (L-chain, molecular weight 25.8 kDa) [38]. The H-chain, the most important component of SF, alternates between crystalline and amorphous regions. The crystalline region is dominated by the hydrophobic sequence Gly-Ala-Gly-Ala-Gly-Ser (GAGAGS) amino acid (Figure 3). The H-chain is organized into 12 crystalline domains interspersed with amorphous regions that enable the protein to transition from random coil and α-helix conformations to antipolar-antiparallel beta-sheet (β-sheets) containing structures. The propensity of the H-chain to form β-sheets under various external stimuli (i.e., temperature, pH, ionic strength, etc.) enables the processing of the protein solution into numerous formats (such as gels, films, sponges, fibers, etc.) with tunable physical and mechanical properties [39,40]. These sequences play a key role in the crystalline structure and mechanical properties of SF. The amino acid residues in the amorphous region mostly have large side chains, such as lysine, tyrosine, and arginine, with non-repetitive sequences that give the molecular chain segments some flexibility [41].

The secondary structure of SF is the relatively stable structure formed by the main chain of the protein molecule through hydrogen bonds. The secondary structure of proteins mainly includes random coil, α-helix, β-sheet, and β-turn, which can transform into each other under certain conditions. The α-helix structure contains more hydrogen bonds and is more stable than the random coil structure. In the β-sheet structure, hydrogen bonds are located between adjacent β-chains (GAAS tetrapeptide) [42]. The β-sheet structure is more stable than the α-helix [43]. Silk fibers have high β-sheet content. The secondary structure of SF solution produced by alkali degumming and chemical reagent dissolution is mainly in the form of a random coil.

The crystalline structures of SF are mainly silk I and silk II [44]. Silk I is a metastable structure with a crankshaft or S-shaped zigzag configuration, belonging to the orthorhombic system. Silk II is an antiparallel β-sheet structure, belonging to the monoclinic system [45]. Additionally, research has shown that there is a silk III crystalline structure at the silk solution-air interface, which belongs to the hexagonal system with a three-fold helix configuration of the peptide chain [46].

**Figure 3 materials-17-03924-f003:**
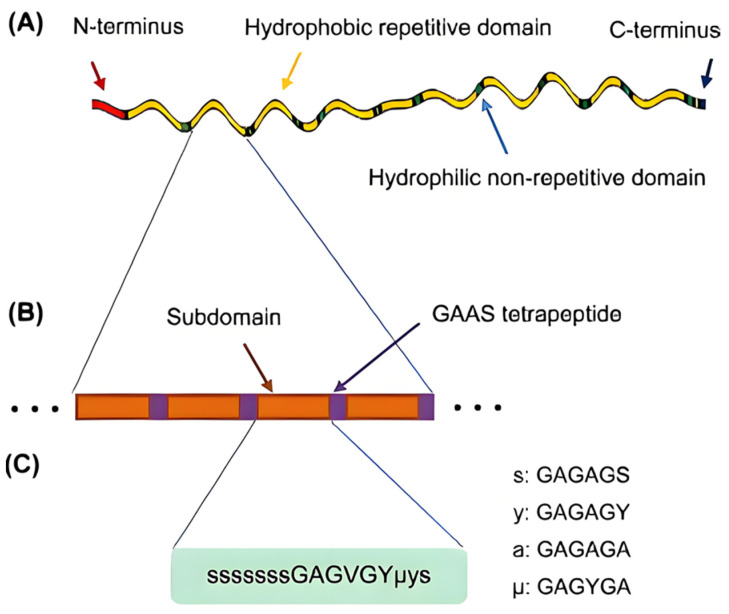
Structural analysis of silk fibroin H-chain (**A**) SF H-chain is composed of hydrophobic domains interspersed among hydrophilic domains. (**B**) The repetitive domain consists of subdomains separated by GAAS tetrapeptides. (**C**) Each subdomain is further composed of different repetitive units of hexapeptides and terminates with the tetrapeptide GAAS [47].

### 3.2. Biological Properties of SF Materials

#### 3.2.1. Immunogenicity

Immunogenicity refers to the ability of an antigen to stimulate specific immune cells, activating, proliferating, and differentiating them, ultimately producing immune effectors (antibodies) and sensitized lymphocytes. When using foreign proteins, the primary concern is their immunogenicity. However, SF has very low immunogenicity, almost not causing an immune response. Studies have shown that SF rarely induces allergic reactions, and most mild immune responses may be caused by immune cells recruited around the silk material, with the main influencing factor being other proteins adsorbed on the SF material [48]. Nonetheless, SF materials need to undergo various treatments before application. Depending on the intended use, they are prepared in different forms (films, hydrogels, scaffolds, particles, fibers, etc.), processed with different techniques (such as ultrasound, casting, or freeze-drying), and sterilized using ultraviolet radiation or other methods. These processing steps affect the mechanical and degradation properties of SF materials and correspondingly influence the interaction between immune cells and SF materials [48].

#### 3.2.2. Cell Compatibility

SF materials support and promote the adhesion, migration, proliferation, and differentiation of various cells such as fibroblasts, endothelial cells, and osteoblasts [49,50,51,52]. Bhardwaj et al. [53] obtained porous 3D SF–keratin scaffolds that had high porosity and swelling ability. The research results showed that the SF keratin complex exhibits strong abilities for fibroblast growth, attachment, and proliferation, and secretes type I collagen in the scaffolds. Stoppato et al. [54] prepared a poly(d,l-lactic acid) (PDLLA) salt-leached sponge and modified it by incorporating SF fibers to form multi-component scaffolds. The results showed that adding SF fibers to the PDLLA salt-leached sponge increased the scaffold stiffness and enhanced its ability to support endothelial cells in vitro and promote angiogenesis in vivo. Platelet endothelial cell adhesion molecular (PECAM-1) mechanosensors at cell–cell junctions indicated better support for endothelial cell growth in static culture on scaffolds containing silk fibers at 21 days. Park et al. [55] developed a 3D electrospun SF (ESF) scaffold with adjustable pore size using a salt leaching method. In this study, the ESF scaffold was compared with a commercially available porous 3D polylactic acid (PLA) scaffold. The results showed high proliferation and osteogenic activity of osteoblasts in ESF scaffolds both in vitro and in vivo.

#### 3.2.3. Blood Compatibility

SF materials have blood compatibility and can promote vascular tissue regeneration and enhance the body’s antithrombotic ability. Compared to other biological protein materials, the risk of SF infection is lower. When applied to wounds, SF can adhere to the wound through dissolution or swelling, providing hemostasis. Additionally, SF can adsorb and concentrate effective components in the blood, such as clotting factors and platelets, to promote the hemostasis process. Catto et al. [56] prepared nanostructured tubular scaffolds (ES-SF) using electrospinning technology. The study showed that these ES-SF scaffolds allowed for the in vitro adhesion and growth of primary pig aortic smooth muscle cells, promoting vascular tissue growth and serving as a scaffold for small-diameter vascular regeneration.

#### 3.2.4. Biodegradability

SF can be biodegraded both in vitro and in vivo, with non-toxic and harmless degradation products and no adverse effects on tissue cells [57,58,59]. Since the human body lacks specific enzymes for SF degradation, the degradation of SF in the body is non-specific. Especially for some crystalline structures of SF, degradation can only proceed gradually from one end of the protein, resulting in a slow degradation rate controlled by the aggregated state structure of SF. Natural silk fibers used as surgical sutures lose their strength in the body after about a year and disappear after two years, indicating that silk can be completely biodegraded in the body. The degradation rate can be controlled by adjusting the aggregated state structure of regenerated SF materials, modulating the porous structure of SF materials, and combining it with other materials. Of course, the degradation rate also depends on factors such as the implantation site, mechanical environment, and the patient’s physiological state. The degradation time of SF can be controlled, ranging from rapid degradation in a few weeks to slow degradation over several years.

#### 3.2.5. In Vivo Implantation

SF induces a mild inflammatory response in the initial stage of implantation, which is beneficial for eliminating pathogens at the injury site and promoting the release of growth factors, aiding wound repair. Panilaitis et al. [60] studied the response of macrophages to silk. It was shown that the low inflammatory potential of silk is based on the release of the pro-inflammatory cytokine tumor necrosis factor-α (TNF-α). The inflammatory response at the initial stage of silk implantation quickly subsides without the risk of developing into chronic inflammation [48]. Additionally, SF can promote the expression of fibronectin and vascular endothelial growth factor in fibroblasts, activate the NF-κB signaling pathway, and induce wound re-epithelialization, thereby facilitating wound healing [61].

#### 3.2.6. Promoting Tissue Repair

Wound dressings made from SF can modulate the cellular expression of pro-inflammatory cytokines (IL-α, IL-6) and anti-inflammatory cytokines (IL-10) by stimulating NF-κB signaling pathways after skin trauma, inhibiting the expression of vimentin, fibronectin, cyclin D1, and vascular endothelial growth factors to promote wound healing [61,62].

## 4. Preparation of Regenerated SF Soft Materials

SF fibers can be conventionally manufactured into woven, knitted, and non-woven fabrics. With its diverse processability, SF solution can be prepared into various forms such as porous scaffolds, hydrogels, films, nanofibers, microspheres, microneedles, etc. [63,64,65] (Figure 4). These different forms of SF materials have been widely used in various fields, including surgical sutures, drug delivery carriers, and tissue engineering scaffolds.

### 4.1. Preparation of SF Films

Compared to other forms, SF film preparation and characterization are simple and widely studied. SF films can be divided into dense films, porous films, and layered films based on their structural form. Dense films are mainly prepared by solution casting or spin coating methods [66,67]. However, compared to natural mulberry silk, the mechanical properties of SF films are often significantly reduced. Therefore, SF is often mixed with other substances to prepare composite SF films for various applications [68].

#### 4.1.1. Control of Aggregate Structure

The preparation of dense films is the basis for the preparation of porous and layered films. During the preparation of dense films using the solution casting method, the drying temperature is the main factor affecting the aggregated structure of SF. Drying at lower temperatures (below 40 °C) results in low β-sheet content in the SF. Higher temperatures (above 60 °C) accelerate the molecular movement of SF, forming more β-sheet structures. Porous films are usually prepared by adding a porogenic agent to the silk solution, followed by solution casting. There are many methods for forming the pore structure of silk films, commonly including freeze-drying and salt leaching [69,70]. The pore size, pore density, and pore distribution all affect the structure of the SF film. Layered films are mainly prepared by spin-coating multiple layers of dense films. The materials used for stacking and appropriate stacking methods are crucial for the performance of the SF film. Post-treatment of the films, such as soaking in small molecule alcohol solvents, reduces the random coil structures in the SF film, increasing the β-sheet structures. Water vapor annealing treatment allows for finer control of the β-sheet content by adjusting the temperature. Studies have found that SF films formed at temperatures above 60 °C and below 95 °C have higher β-sheet content and stronger interlayer interactions [71,72].

#### 4.1.2. Chemical Crosslinking

Chemically crosslinked SF films help meet the stringent mechanical performance requirements for practical applications. Common chemical crosslinking agents for SF film modification include epoxy resin, glyoxal, glutaraldehyde, genipin, dialdehyde starch, and 1-ethyl-3-(3-dimethylaminopropyl) carbodiimide/N-hydroxysuccinimide (EDC/NHS), among others. Yang et al. [73] used epoxy resin to modify SF films. The results showed that the original hydrogen bond interactions between SF chains were disrupted, allowing the chains to slide against each other. The prepared SF films exhibited higher elongation at break. On the other hand, the SF formed a crosslinked network structure, enhancing the recoverability after chain slippage, resulting in good elasticity and increased strength. Kharlampieva et al. [74] used glutaraldehyde for SF crosslinking and then mixed it with nano-silica to construct a highly crosslinked silk-based nanocomposite film in one step. Its elastic modulus and tensile strength increased several times compared to conventional SF films. Attenuated total reflectance Fourier-transform infrared (ATR-FTIR) structural analysis revealed that the β-sheet content of SF increased after the introduction of glutaraldehyde as a crosslinking agent.

#### 4.1.3. Preparation of Composite Films

The dense, non-porous structure, low wet film strength, and poor support of SF films limit their application range. Mixing SF with polymers to prepare composite films can effectively address these shortcomings. Suzuki et al. [75] mixed SF solution with polyethylene glycol, crosslinked with genipin, to prepare blended SF materials. Compared with untreated films, the mechanical properties of the blended materials were significantly improved. This blended film showed good cell compatibility in the primary culture of human corneal limbal epithelial cells. Ma et al. [76] prepared Poly l(+) lactic acid/SF (PLLA/SF) fiber films using coaxial electrospinning. The results showed that the addition of SF increased the drug release efficiency in the fibers. Shao et al. [77] electrospun PLA/SF fiber films with significantly improved mechanical properties. The composite film promoted osteoblast differentiation and could be used in bone tissue engineering. Song et al. [78] added SF to electrospun porous PLLA fiber films, improving the material’s hydrophilicity and cell compatibility.

### 4.2. Hydrogels

Hydrogels have a three-dimensional network structure containing a significant amount of water and porous structures. The formation of SF hydrogels depends on the density of intermolecular crosslinking points. Hydrogels can be formed through physical crosslinking or chemical crosslinking methods.

#### 4.2.1. Physical Crosslinking

Physical crosslinking is achieved by changing physical conditions to promote strong hydrogen bonding crosslinking between SF molecules, mainly forming β-sheet structures. Sufficient crosslinking points constituted by β-sheet structures ultimately complete the sol–gel transition. For protein-based materials, temperature is a key factor in gel formation. In temperature-triggered gelation, rising temperatures promote the frequency of molecular collisions in the system, enhancing the aggregation of SF molecules. Additionally, the hydrophobic segments of SF increase with rising temperatures, strengthening hydrophobic interactions between molecules [79,80,81]. These factors collectively lead to the aggregation and assembly of SF. In SF, the β-sheet structures formed have thermodynamic stability, making the sol–gel transition triggered by high temperatures irreversible. Generally, SF solution can be stored at 4 °C for several weeks. However, placing the SF solution at room temperature will quickly result in gelation [82]. Shear gelation is achieved by subjecting SF solution to high-speed vortex shear treatment to form gels. The mechanism is that shear forces cause fluid rotation and stretching, leading to the stretching and alignment of SF macromolecules, and promoting their aggregation [83]. With increasing vortex time, the β-sheet content in SF increases, and the crosslinking density gradually rises, promoting the gelation of SF. Therefore, the SF solution should avoid high-speed stirring to prevent shear-induced gelation. Ultrasound can affect other physical factors, such as local temperature rise, shear force extension, and changes in gas–liquid interface balance. Ultrasonic waves can promote the rapid gelation of SF solution [84]. In this method, ultrasonic power output, ultrasonic duration, ultrasonic output power, and SF concentration are important factors in regulating the state of hydrogels.

#### 4.2.2. Chemical Crosslinking

The charge of SF affects the interactions between SF and other molecules. When the pH of the SF solution is equal to its isoelectric point (PI), it is likely to aggregate into a hydrogel state [85]. Adding acidic substances to the SF solution to adjust the pH near the isoelectric point can induce SF gelation. An electric field can also induce the gelation of SF solution. Under the influence of an electric field, a large number of protons in the solution move toward the anode, leading to a local pH decrease near the isoelectric point of SF [86]. SF molecules aggregate near the anode and form a hydrogel. Water-soluble organic solvents such as methanol and ethanol can also induce SF solution to form hydrogels. The addition of ethanol dehydrates the hydrophobic domains of SF, bringing them closer together to form hydrogen bonds and β-sheet structured crosslinking points. Subsequently, the β-sheet structured crosslinking points link together to form a network structure, causing the SF solution to hydrogel [87]. Adding surfactants can induce rapid gelation of SF solution [88]. This process is mainly based on the amphiphilicity of surfactants, allowing them to encapsulate the hydrophobic segments of SF, reducing the surface tension of the SF solution and forming hydrogen bonds with SF, promoting the formation of β-sheet structures.

Photopolymerized hydrogels are formed through photoinitiated polymerization reactions in the presence of photoinitiators under visible or ultraviolet light. The photoinitiator molecules absorb ultraviolet light energy, entering an excited state and generating free radicals upon cleavage. These free radicals subsequently initiate the unsaturated bonds (vinyl bonds) in the monomer molecules, leading to free radical polymerization and eventually crosslinking to form hydrogels [89]. Radiation can generate free radicals between unsaturated polymer chains and water molecules, inducing intermolecular crosslinking [90]. Its advantage is that the preparation of SF hydrogels does not require potentially toxic initiators and crosslinkers. Enzyme crosslinking reactions mainly utilize the catalytic action of biological enzymes to activate specific groups on the side chains of SF, promoting chemical crosslinking between molecular chains. This method features mild reaction conditions, avoiding the introduction of toxic crosslinkers and organic solvents. Additionally, it offers good biocompatibility and moderate reaction conditions. Commonly used enzymes include horseradish peroxidase [91], tyrosinase [92], and laccase [93], among others. Natural polymers exhibit higher biocompatibility, excellent biodegradability, and non-toxicity. To date, collagen, gelatin, chitosan, cellulose, alginate, starch, SF, and HA, either alone or in combination, have been widely used in tissue engineering. The combination of SF with cellulose, alginate, and chitosan has shown outstanding capabilities in tissue regeneration engineering, wound healing, and drug delivery [94]. SF can also be mixed with synthetic polymers. After blending, the properties of the macromolecular polymers can be adjusted, altering the mechanical properties and degradability of the polymers to meet the needs of different tissue regeneration applications [95].

In summary, physical crosslinking methods are relatively simple, involving no intricate steps or chemical reagents. However, hydrogels prepared through physical crosslinking tend to be brittle with inferior mechanical properties. Conversely, SF hydrogels prepared via chemical crosslinking exhibit stable structures and superior mechanical performance. Furthermore, chemical crosslinking effectively modifies the hydrogel’s pore size, swelling properties, and biodegradability. Nonetheless, photo-initiators and chemical crosslinking agents possess cytotoxicity, rendering them unsuitable for direct biomedical applications. Enzymatic and radiation crosslinking methods are safer alternatives compared to photo-initiators and chemical crosslinking agents. Therefore, to fabricate hydrogel materials with excellent performance, a combination of multiple methods is advisable. The diversity in crosslinking methods enriches the network structure of hydrogels, thereby enhancing the multifunctionality of SF hydrogels (Figure 5).

### 4.3. Porous Scaffolds

Porous scaffolds exhibit suitable mechanical properties and contain numerous internal pores that facilitate cell adhesion, proliferation, and migration. The presence of these pores supports the diffusion of nutrients and the removal of metabolic waste, making them promising biomaterials. Various methods currently exist for preparing porous scaffolds based on SF, such as freeze-drying, particle leaching, gas foaming, supercritical CO_2_ treatment, and 3D bioprinting (Figure 6) [97].

#### 4.3.1. Freeze-Drying

Freeze-drying is the primary technique used to prepare SF-based porous scaffolds. By inducing the sublimation of ice crystals in the frozen SF aqueous solution under vacuum, pores are generated within the scaffold. The characteristics of these pores, including their shape, quantity, and size, can be controlled by adjusting the freezing time, temperature, and SF concentration [95]. SF can be directly used or doped with other materials (such as collagen or hyaluronic acid) to form three-dimensional scaffolds via EDC/NHS crosslinking and freeze-drying methods [98,99]. The freeze-drying method is simple, and the preparation process generally does not involve organic solvents, ensuring the good biocompatibility of the scaffolds and excellent connectivity between pores. However, scaffolds manufactured using the freeze-drying method typically exhibit smaller pore sizes and weaker mechanical properties.

#### 4.3.2. Particle Leaching

Particle leaching is another commonly used method in tissue engineering to prepare scaffolds. This involves using porogens such as salt, sugar, or wax to form pores or channels. By adjusting the size, shape, or quantity of the porogens used, one can control the pore size. Nevertheless, this method has limited capability in adjusting pore openings, shapes, or the mechanical properties of the scaffold. Sodium chloride (NaCl) particles are currently the most widely used porogens. Xiao et al. [100,101] developed an SF scaffold with nanofiber microporous structures and fewer β-sheet structures through salt leaching. The mechanical properties of this scaffold can be adjusted by modulating the interactions between SF and water.

#### 4.3.3. Gas Foaming

Gas foaming methods primarily include physical foaming and chemical foaming. Physical foaming involves adding H_2_, CO_2_, and other inert gases to induce bubble formation, whereas chemical foaming uses chemical processes (such as the decomposition of NH_4_HCO_3_ and NaBH_4_) to form bubbles in situ [102]. Although scaffolds obtained through gas foaming usually have high porosity avoid the use of organic solvents, are cost-effective, and have short processing times, the method cannot precisely control pore distribution and porosity.

#### 4.3.4. Supercritical CO_2_ Processing

Supercritical CO_2_ technology exposes materials to conditions exceeding critical pressure and temperature, generating supercritical CO_2_ fluid. Under these circumstances, the gaseous and liquid phases of the material become indistinguishable. This non-solvent-based method can be used to customize the internal porous structure of scaffolds [103]. The technology avoids the use of organic solvents and leverages the advantages of supercritical gas, such as low temperature, non-toxicity, fast mass transfer, and the absence of residual substances, thereby significantly improving the previous gas foaming method’s requirement for high temperatures during material preparation. Additionally, under such experimental conditions, when active factors form complexes with the scaffold, their activity can be well maintained.

#### 4.3.5. Three-Dimensional Bioprinting

Three-dimensional bioprinting includes a series of printing technologies such as inkjet bioprinting, laser-assisted bioprinting, microextrusion bioprinting, and two-photon polymerization-based bioprinting. Materials suitable for 3D printing must have appropriate rheological properties to meet the demands of the printing process and solidification properties to ensure the formation of mechanically stable structures. Despite SF exhibiting good shear-thinning and viscosity characteristics for 3D printing, its slow gelation rate and relatively harsh gel formation conditions often necessitate printing it alongside another polymeric material [104].

#### 4.3.6. Electrospinning Nonwoven Fabric

During the electrospinning process [105], an SF solution is fed through a conical nozzle via a conduit and extruded slowly. Under the stretching forces of a high electric field, the surface tension of the spinning solution is overcome, forming jets. These jets are refined under the electric field’s stretching forces, ultimately forming nanofibers that deposit on a collecting plate, resulting in the production of electrospun nonwoven fabric with nanopores.

## 5. Applications of SF Soft Materials

### 5.1. Applications of SF Hydrogels

SF hydrogels have extensive applications in the biomedical field. The biochemical composition and properties of hydrogels, such as water content, viscoelasticity, and mechanical strength, are similar to natural tissues [106]. Hydrogels have the capability to transport biologically active molecules, such as growth factors, hormones, and peptide sequences, while maintaining structural integrity, making them applicable in tissue repair [107]. Another advantage of using hydrogels in tissue repair is their mild preparation conditions, which do not affect cell viability. During the production process, cells can be encapsulated within the hydrogel, increasing the proliferation of cell populations near the gel surface during application [108]. Consequently, hydrogels serve as versatile carriers with substantial applications in biological tissue engineering.

SF hydrogels can be employed as bioactive materials implanted in damaged bone or cartilage tissues to induce bone formation and ultimately lead to bone healing, displaying superior capabilities in promoting cell metabolism and bone remodeling. Studies have shown that SF hydrogels, induced by acetic acid, can form injectable hydrogels. In vitro cell experiments revealed that these hydrogels promote the proliferation of MG63 osteoblasts and stimulate their secretion of more TGF-β1. When injected in vivo, these hydrogels repaired critical-sized defects in the trabecular bone of rabbits and significantly improved bone remodeling and maturation. Overall, bone healing rates, as well as the proliferation and differentiation of osteoblasts in the presence of SF hydrogels, were superior to control groups [109]. However, due to the poor mechanical properties of pure SF hydrogels, they can only be used for repairing non-load-bearing, irregular bone defects or cartilage injuries. To meet the requirements of bone tissue repair, SF is typically combined with other materials to prepare composite SF hydrogels. Hydroxyapatite (HAp), the primary inorganic component of bone tissue, possesses good biocompatibility and osteoconductivity. Composite materials of HAp and SF form ideal bone repair materials. Bai et al. [110] used SF as a polymer to connect host (β-cyclodextrin) and guest (cholesterol) bodies. Due to the dynamic host–guest interactions, composite SF hydrogels prepared by this method can self-repair when damaged, effectively mimicking the self-healing properties of natural bone tissue. Additionally, the host–guest crosslinking confers strong mechanical properties to the hydrogels, enabling them to withstand high mechanical loads. HA was also incorporated into these hydrogels. Cell and animal experiments showed that the composite hydrogels effectively promoted cell proliferation and osteoblastic differentiation, accelerating bone regeneration in critical-sized femoral defects in rats. Zheng et al. [111] combined methacryloyl gelatin and SF under UV irradiation and ethanol treatment to form composite hydrogels with interpenetrating network structures. Compared to pure SF hydrogels, these composite hydrogels significantly increased mechanical strength, achieving a compressive modulus of 300 kPa. In vitro studies have revealed that the formation of interpenetrating network structures in composite hydrogels did not compromise their biocompatibility or cell adhesion properties. When bone mesenchymal stem cells(BMSC) were seeded onto the hydrogels, in vivo experimental results showed that cartilage was only partially regenerated in the group treated with hydrogels alone. However, cartilage repair was superior in the group treated with cell-seeded hydrogels, indicating that additional cell seeding is required in hydrogel-based cartilage treatments (Figure 7).

### 5.2. Applications of SF Porous Scaffolds

The porous structure of SF scaffolds provides ample space for cell adhesion, proliferation, and migration. Additionally, nutrients and moisture can be transported directly to cells through these voids. The formation of the extracellular matrix, growth of new tissue, and transport of metabolic waste can all be facilitated by the porous structure. Leveraging these characteristics, SF porous scaffolds have been widely applied in various fields including skin, bone, cartilage, blood vessels, ligaments, nerves, tendons, and others [97].

#### 5.2.1. Skin Tissue Regeneration

Yan et al. [113] prepared an SF/chondroitin sulfate (CS)/HA ternary scaffold using freeze-drying. By combining CS and HA with SF solution, they controlled the chemical potential and water content around ice crystals to form smaller pores within the SF/CS/HA ternary scaffold’s main pores, thereby improving the scaffold’s equilibrium swelling. This feature benefits cell adhesion, survival, and proliferation. Results from in vivo experiments on full-thickness wounds on the backs of rats showed that the SF/CS/HA ternary scaffold promoted dermal regeneration, improved vascularization, and collagen deposition. Furthermore, the expression and secretion of vascular endothelial growth factor (VEGF), platelet-derived growth factor (PDGF), and basic fibroblast growth factor (bFGF) in the SF/CS/HA ternary scaffold accelerated the wound healing process. These SF/CS/HA ternary scaffolds hold the potential for dermal regeneration.

SF scaffolds loaded with bioactive components, cytokines, cells, and tissues not only provide physical support but also act as delivery systems for wound repair. Xie et al. [114] developed an SF-based scaffold for delivering stem cells into burn wounds in rats. SF scaffolds significantly accelerated collagen synthesis and re-epithelialization of the skin. The histological characteristics of the reconstructed skin and its appendages resembled those of normal rat skin. Additionally, collagen/SF hybrid scaffolds loaded with bone marrow mesenchymal stem cells demonstrated excellent skin affinity, breathability, and water permeability [115].

#### 5.2.2. Bone Tissue Regeneration

SF scaffolds also play a crucial role in bone tissue regeneration due to their excellent biocompatibility, favorable for cell adhesion, growth, differentiation, migration, and promoting osteogenesis and oxygen transport capabilities. Riccio et al. [116] found that SF scaffolds repaired cranial defects irrespective of whether human stem cells were pre-seeded into them or not. Higher levels of osteogenesis were observed in the SF scaffold experimental group pre-implanted with stem cells. Wu et al. [117] manufactured PLLA/SF composite nanofiber scaffolds using electrospinning and coated them with osteoblast-derived extracellular matrix (O-ECM) on the scaffold. In vitro experiments demonstrated that the novel nanofiber scaffold (O-ECM/PLLA/SF) significantly enhanced the osteogenic differentiation ability of cultured stem cells. Compared to pure alginate and alginate/HAp, alginate/HAp/SF composite materials exhibited significantly higher levels of new bone formation and reduced TNF-α levels [118]. In a 3D porous HAp/SF/sodium alginate scaffold, a higher SF/HAp to sodium alginate ratio improved cell proliferation and enhanced alkaline phosphatase activity. In another study, graphene oxide-modified SF/HAp scaffolds loaded with stem cells promoted bone regeneration and immunomodulatory effects [119].

#### 5.2.3. Cartilage Regeneration

SF has been studied for decades for repairing cartilage. Aoki et al. [120] confirmed the proliferation and differentiation phenotype of chondrocytes in SF sponges. In SF-based scaffolds used for cartilage tissue regeneration, pore size, and porosity significantly affect cell attachment and infiltration. Pores smaller than 300 μm contribute to chondrogenesis, while pores larger than 300 μm aid in osteogenesis [121]. Wuttisiriboon et al. [122] prepared a composite scaffold consisting of SF, gelatin, chondroitin sulfate, hyaluronic acid, and aloe vera through freeze-drying. The scaffold had an interconnected porous structure with an average pore size of approximately 209 μm. Additionally, it exhibited a high absorption rate and good mechanical strength; moreover, it retained its structure for up to 21 days. Cell experiments showed that human bone marrow mesenchymal stem cells (BM-MSCs) proliferated more rapidly when using this scaffold compared to using a pure SF scaffold.

#### 5.2.4. Vascular Tissue Regeneration

Studies have demonstrated that SF scaffolds support the adhesion, growth, survival, and proliferation of three types of vascular cells: human aortic smooth muscle cells, human coronary artery endothelial cells, and human aortic adventitial fibroblasts cells. SF-based vascular grafts tend to develop a thin luminal layer, facilitating rapid endothelialization [123,124,125]. The ability of vessels to grow within SF scaffolds varies with the morphology of the SF scaffold. Diameter and porosity are common influencing factors affecting cell infiltration, adhesion, and proliferation behavior [126]. Sun et al. [127] manufactured SF tubular scaffolds with different pore sizes. They found that micropores of 30–50 µm were suitable for the proliferation and growth of human umbilical vein endothelial cells (HUVEC). Yang et al. [128] developed a composite scaffold of SF and fibronectin using electrospinning technology to simulate natural blood vessels. The scaffold exhibited a smooth and uniform fiber structure with smaller fiber diameters, showing excellent blood compatibility and an appropriate biodegradation rate. In addition, it increases the adhesion and proliferation of MSCs. These results make it a potential material for artificial vascular stents. Asakura et al. [129] studied the application of SF materials in vascular repair. They found that SF has a unique reshaping function compared to polyester fibers or expanded PTFE grafts. They coated woven SF grafts with SF solution and crosslinking agent poly(ethylene glycol diglycidyl ether), producing small diameter vascular grafts with diameters of 1.5 mm and lengths of 10 mm. The grafts exhibited excellent physical strength, while the coating on them prevented blood leakage and increased elasticity.

#### 5.2.5. Ligament and Tendon Regeneration

An important component of the knee joint is the anterior cruciate ligament (ACL). Improper movements and excessive external forces can lead to ACL injury, resulting in knee instability and progressive damage. Artificial ligaments can reduce the risk of donor site morbidity or disease transmission associated with autografts or allografts. SF has been shown to support the differentiation of adult stem cells into ligament lineages [52]. Geng et al. [130] prepared an SF/collagen porous scaffold characterized by a compositional gradient that mimics the natural tendon structure. It exhibited good compatibility in cell experiments and promoted tendon regeneration. Another method for ligament regeneration is to seed cells into an SF-based scaffold prior to implantation to guide ligament–bone insertion. Ribeiro et al. [131] studied a biomimetic composite scaffold composed of SF hydrogel crosslinked by horseradish peroxidase containing ZnSr-doped β-tricalcium phosphate particles. The scaffold exhibited sufficient structural integrity, swelling capacity, and tensile strength. After 14 days of in vitro culture, it demonstrated vitality in cell adhesion and proliferation.

In clinical practice, a commercial product used for posterior cruciate ligament replacement is the Ligament Augmentation and Reconstruction System (LARS), composed of polyethylene terephthalate (PET). However, LARS has disadvantages such as complications like joint fibrosis and heterotopic ossification [132,133]. Jiang et al. [134] modified PET surfaces with SF to alter their hydrophilicity and biocompatibility. A series of in vitro experiments confirmed that the SF coating enhanced cell adhesion and proliferation, improving the material’s biocompatibility and its “ligamentization” process. SF can compensate for the deficiencies of PET, inducing autologous tissue ingrowth. A three-layer coated scaffold was introduced on the surface of PET artificial ligaments using a stepwise deposition method, incorporating heparin and bone morphogenetic protein-binding peptides. This three-layer coated scaffold not only promoted the biocompatibility of PET grafts but also regulated early inflammatory responses in the joint cavity, promoting and improving graft bone integration, showing enormous potential in enhancing ACL reconstruction clinical efficacy [135].

#### 5.2.6. Nerve Tissue Regeneration

The treatment of nerve defects poses a significant challenge. Autologous nerve transplantation is one therapeutic approach. However, it is limited by the availability of autologous donor tissues and issues such as reduced donor site sensitivity, adhesive scars, and neurofibroma formation [136]. Many studies have combined SF with other materials such as poly(lactic-co-glycolic) acid (PLGA), polypropylene (PPY), polyethylene oxide, and collagen to explore better materials for nerve defect repair. Tang et al. [137] co-cultured dorsal root ganglia and Schwann cells on SF-based scaffolds to form neuroequivalents of nerve grafts in vitro. Twelve weeks after nerve transplantation, the graft induced better nerve regeneration and functional recovery than pure SF scaffolds. Xue et al. [138] studied electrospun SF conduits for bridging a 30 mm sciatic nerve gap in dogs. Histological and functional evaluations after 12 months showed that SF-based nerve scaffolds had acceptable regeneration outcomes comparable to autograft groups.

SF three-dimensional scaffolds can promote neural tissue regeneration by providing oriented structural support and guidance for axonal growth. Future research may focus on improving the three-dimensional spatial structure of scaffold structures and incorporating neurotrophic factors to enhance axonal regeneration, especially in cases of spinal cord injury or peripheral nerve injury.

#### 5.2.7. Regeneration of Other Tissues

SF has also been explored for the repair of less common tissues, such as dental, gastrointestinal, and urethral tissues. Lopez-Garcia et al. [139] found that SF three-dimensional scaffolds coated with graphene can differentiate human dental pulp stem cells by promoting the mineralization of extracellular matrix. Algarrahi et al. [140] conducted an experiment in a rat esophageal reconstruction model, indicating that SF scaffolds, when used as acellular grafts, resulted in less inflammation and fibrosis compared to traditional small intestinal submucosa implants. Niu et al. [141] successfully fabricated a biomimetic tubular HA-SF nanofiber scaffold through electrospinning and crosslinking processes. This scaffold’s structure, morphology, and mechanical properties closely resemble those of natural rabbit urethral tissue. The nanofiber surface of this scaffold is more suitable for the growth of UC, forming new urothelial tissue.

## 6. Microenvironment Control in Wound Healing

### 6.1. Hemostasis Control in the First Phase of Wound Healing

Adding SF to hemostatic agents can increase coagulation activity and reduce bleeding time, thereby minimizing blood loss [142]. A combination of active (thrombin) and passive (gelatin, cellulose, collagen, chitosan) hemostatic agents is a better choice for hemostatic materials, ensuring bleeding control in the shortest possible time [143,144]. Shefa et al. [145] found that a combination of oxidized cellulose and SF is a good candidate for wound healing. Loading thrombin onto oxidized cellulose/SF scaffolds can significantly improve hemostatic performance. Sabarees et al. [146] evaluated the potential of chitosan hydrogel (CHI-HYD) and chitosan/SF hydrogel (CHI-SF-HYD) in blood coagulation and wound healing. Coagulation experiments showed that the composite hydrogel exhibited hemostatic activity. The blood coagulation time for CHI-HYD and CHI-SF-HYD was shorter than the natural coagulation process.

### 6.2. Antibacterial Control in the Second Phase of Wound Healing

Bacterial contamination in the wound healing process is a prevalent issue, posing a serious threat to global human health. Therefore, developing multifunctional biomaterials with antibacterial properties and infection resistance is a continuous goal for biomedical applications. SF has excellent mechanical properties, high biocompatibility, and good biodegradability [147]. Due to the widespread application of SF-based biomaterials, many researchers over the past decade have explored different strategies to impart antibacterial properties to these materials.

Antibiotics are among the most commonly used drugs in combination with SF to prevent and treat bacterial infections. Systemic administration of antibiotics may lead to many adverse side effects. The solution is to use implantable biomaterials to allow antibiotics to be slowly released in the scaffold, achieving higher doses and fewer side effects through local release at the site of infection [148]. Lan et al. [149] prepared gelatin microspheres loaded with vancomycin and embedded them into freeze-dried SF scaffolds for wound repair. Pritchard et al. [150] studied the loading and release of various antibiotics in different forms of SF materials, including films, microspheres, hydrogels, and coatings. Gentamicin loaded on silk sponges had a release time of 5 days, while cefazolin had a release time of 3 days. Additionally, SF hydrogels can be easily applied to wounds and removed without causing discomfort or additional injury to patients [151].

In recent years, increasing research has focused on developing SF scaffolds with antibacterial properties for wound dressings. Zhang et al. [152] found that incorporating Pluronic polymers into SF can optimize its mechanical properties, hydrophilicity, and light transmittance. The resulting SF scaffolds can be used to encapsulate antimicrobial agents such as curcumin, Ag nanoparticles, and antimicrobial peptide KR-12. Furthermore, in vitro studies showed that SF scaffolds can release antimicrobial agents continuously, killing bacteria. In vivo tests indicated that these scaffolds not only cleared methicillin-resistant Staphylococcus aureus from the wound area and reduced inflammation but also promoted angiogenesis and epithelial reformation, accelerating the healing of infected wounds.

### 6.3. Promoting Cell Proliferation Control in the Third Phase of Wound Healing

The proliferation phase begins approximately 4–5 days after injury. It includes processes such as angiogenesis, granulation tissue formation ECM formation, and re-epithelialization. Many experimental and clinical reports suggest that oxygen plays a crucial role in wound healing. Sen et al. [153] indicated that, under high oxygen conditions, increased local ROS can induce a higher degree of angiogenesis. Kan et al. [154] conducted in vitro studies on human fibroblasts, finding that collagen content in wounds decreased due to increased synthesis of protease (MMP-1). Moreover, Kang et al. [155] conducted another in vitro study, revealing that daily hyperbaric oxygen treatment at 2.0 atmos (ATM) can stimulate fibroblast proliferation.

During the proliferation phase of wound healing, the polarization of macrophages from the classically activated M1 phenotype to the alternatively activated M2 phenotype marks the transition from the inflammatory phase to the proliferation phase [156]. M2 macrophages can be further subdivided into three different phenotypes (M2a, M2b, and M2c), participating in inflammation inhibition, re-epithelialization, and angiogenesis during the cell proliferation phase. In the later stages of the proliferation phase, macrophages also promote the regression of new capillaries by releasing thrombospondin-1 (TSP-1) to prevent excessive angiogenesis [157]. When it comes to interactions between cells and materials, SF scaffolds allow for cell attachment, proliferation, migration, and differentiation. SF scaffolds stimulate cell migration by activating c-Jun N-terminal kinase (JNK), methyl ethyl ketone (MEK), and phosphatidylinositol kinase (PI3K) signaling pathways. Chouhan et al. [158] demonstrated that silk can assist in cell migration, recruiting them to the wound site. The study showed that SF influences cell migration and promotes healing by regulating the expression of cyclin D1, vimentin, vascular endothelial growth factor (VEGF), and fibronectin, known markers of cell proliferation. Furthermore, Infanger et al. [159] confirmed that SF can promote wound healing by modulating protein expression involved in the proliferation and remodeling stages of the NF-κB signaling pathway.

Wu et al. [117] designed a Rat Bone Marrow Mesenchymal Stem Cells (Rb1)/tranSForming Growth Factor-β(TGF-β1)-loaded SF gelatin porous scaffold (GSTR). This scaffold creates a microenvironment conducive to cartilage regeneration, promoting chondrogenesis in vivo, reducing inflammation levels, and enhancing hyaline cartilage regeneration in vitro. Chen et al. [160] synthesized an SF scaffold loaded with tanshinone IIA (TAN), which can enhance the transcription of genes related to chondrocyte activity, reduce oxidative stress, and thereby promote cartilage regeneration. Maity et al. [161] developed a composite hydrogel that stimulates fibroblast migration in vitro and controls oxidative stress for rapid healing of diabetic wounds.

### 6.4. Control of Tissue Remodeling in the Fourth Phase of Wound Healing

The wound remodeling phase occurs 2–3 weeks after injury. Fibroblasts are the primary cell type responsible for the maturation of the ECM in wounds. Hyaluronic acid, fibronectin, and proteoglycans replace the initial fibrin clot and form collagen fibers during the later stages of repair. Proteoglycans help construct mature, crosslinked collagen fibrils and serve as conduits for cell migration. Despite the peak in collagen content during this phase, the wound strength is only about 30% of normal skin, and, after three months, it reaches approximately 80% of the strength of normal skin [162].

Bhar et al. [163] developed a biomimetic hydrogel scaffold using a decellularized extracellular matrix (dECM) derived from omental tissue and SF. They found that the presence of dECM components in the composite material promoted wound closure, granulation tissue formation, and increased re-epithelialization rate by stimulating angiogenesis. Rivero et al. [164] isolated human Wharton’s jelly mesenchymal stem cells (Wj-MSCs) using an explant method and discovered that a combined treatment involving the injection of Wj-MSCs at wound edges exhibited superior wound healing capabilities compared to single treatments and cell-free SF scaffolds. The use of Wj-MSC-based SF constructs in cell therapy contributed to the generation of high-quality, well-vascularized granulation tissue, enhanced wound re-epithelialization, and reduced fibrotic scar formation by decreasing myofibroblast proliferation.

### 6.5. Composite Control across Different Stages

Wound healing involves a series of complex and overlapping processes that engage various cells and biomolecules. The application of SF in tissue engineering and regenerative medicine is widespread. During normal wound healing, SF influences multiple mediators, such as VEGF, EGF, TGF, interleukin-10 (IL-10), interleukin-1β (IL-1β), fibronectin, vimentin, and cyclin D1 by modulating NF-κB signal transduction [53]. Numerous experiments have highlighted its potential application as a wound dressing in the biomedical field by assessing cell viability, macrophage response, and angiogenesis capabilities (Figure 8).

Wang et al. [165] fabricated a bioactive SF scaffold with nanostructured textures to mimic the chemical and biophysical properties of the ECM. In vivo experimental studies demonstrated that nanostructured fibrin-based scaffolds not only significantly accelerated wound healing but also regulated collagen alignment to prevent scar formation. Ding et al. [166] developed an injectable desferrioxamine (DFO)-loaded SF fiber hydrogel. The study results indicated that the hydrogel promoted angiogenesis in diabetic lesions and reduced chronic inflammation, thereby accelerating the healing process and enhancing collagen deposition. In recent studies, SF has been used to enhance the re-epithelialization, adhesion, and proliferation of human fibroblasts, keratinocytes, and endothelial cells in sponges, gels, and microspheres, showing significant effects in wound healing applications [167].

**Figure 8 materials-17-03924-f008:**
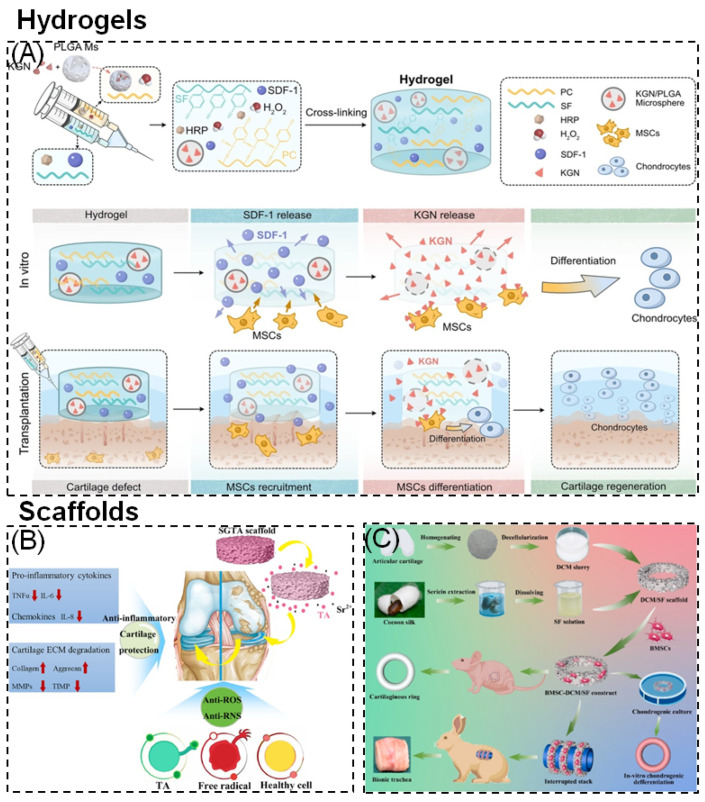
Representative types of SF-based biomaterials [168]: Hydrogels: (**A**) Injectable PC-SF hydrogels loaded with SDF-1, PLGA, and KGN can promote the recruitment and differentiation of stem cells for cartilage regeneration. Scaffolds: (**B**) The mechanism of SF/GO scaffolds can delay osteoarthritis (OA) and protect cartilage. (**C**) Fabrication of DCM/SF scaffold for repairing cartilaginous tissue.

## 7. Summary and Outlook

In the process of wound healing, biomaterials need to provide a suitable environment for the wound and protect it from bacterial infection. SF-based soft materials exhibit the ability to create a moist microenvironment, support cell growth, possess antibacterial properties, and stimulate wound healing at the molecular level. Moreover, they demonstrate good biocompatibility, indicating their potential application as wound healing materials. Currently, research on SF-based wound healing materials is still in its early stages. Despite the wide variety of SF-based materials with different functionalities, an ideal material that integrates multiple functional advantages has not yet been developed. Therefore, this article only introduces silk fibroin membrane, hydrogel, and porous scaffold, and does not introduce a variety of composite materials. In addition, this article does not review the details of the four stages of wound healing. Future research on SF-based soft materials should focus on combining multifunctional properties and the control of various stages of the wound healing process to develop ideal SF-based soft materials for wound repair.

## Figures and Tables

**Figure 1 materials-17-03924-f001:**
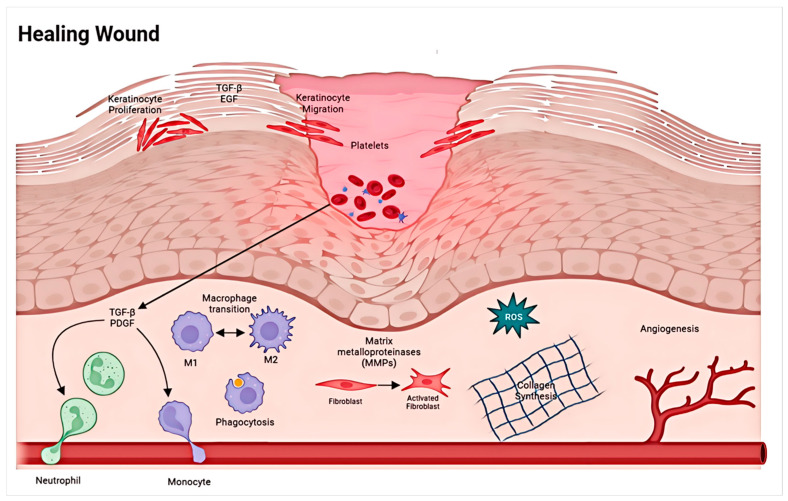
In normal wounds, there is an orderly progression from hemostasis to inflammation, proliferation/repair, and, finally, remodeling [14].

**Figure 2 materials-17-03924-f002:**
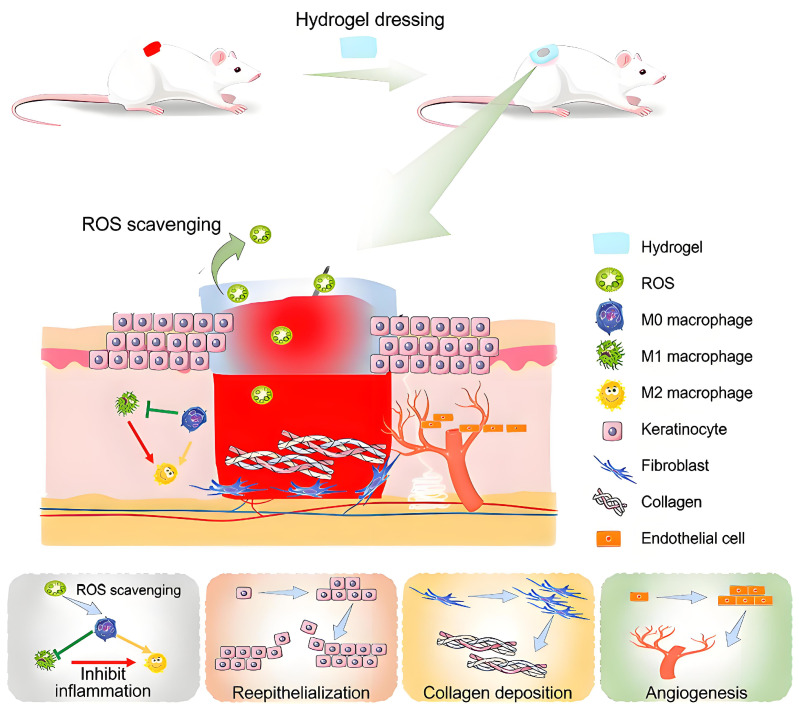
Schematic diagram of hydrogel materials promoting wound healing [20].

**Figure 4 materials-17-03924-f004:**
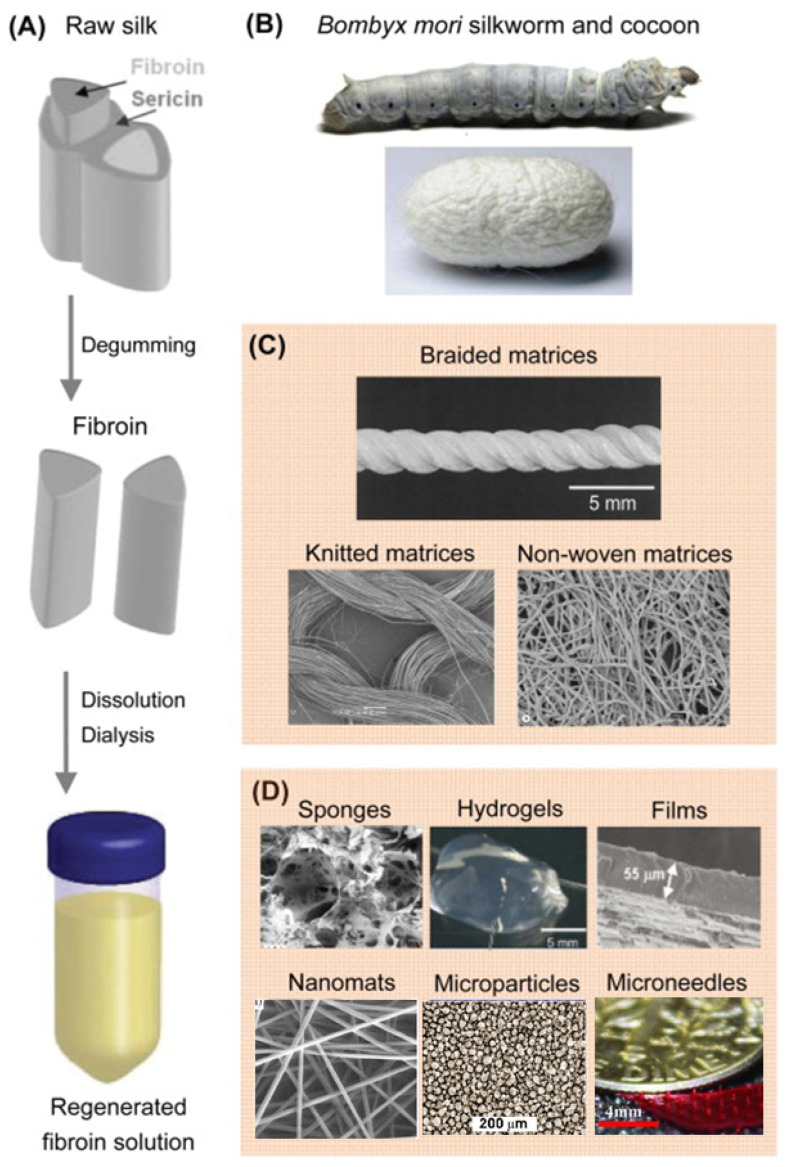
Various forms of SF materials [47]: (**A**) Raw silk consists of two SF fibers coated with sericin. After degumming to remove the sericin, the SF fibers are dissolved in lithium bromide solution, followed by dialysis to obtain a regenerated SF solution. (**B**) Mature silkworms and the cocoons they produce. (**C**) Silk-woven, knitted, and non-woven fabric matrices are composed of SF fibers. (**D**) Sponge, hydrogel, film, nanofibers, microparticles, and microneedles constructed from regenerated SF solution.

**Figure 5 materials-17-03924-f005:**
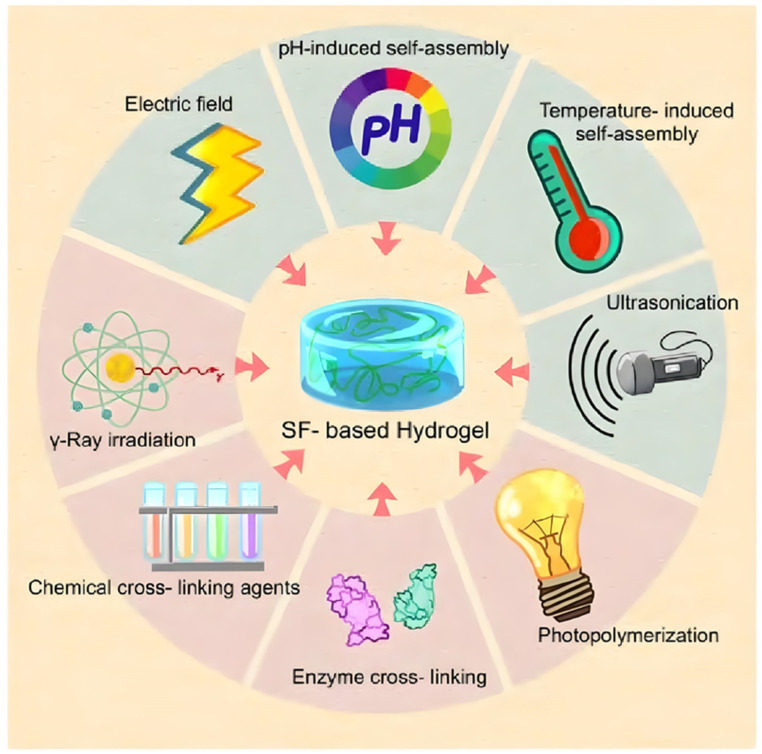
Preparation method of SF hydrogel [96].

**Figure 6 materials-17-03924-f006:**
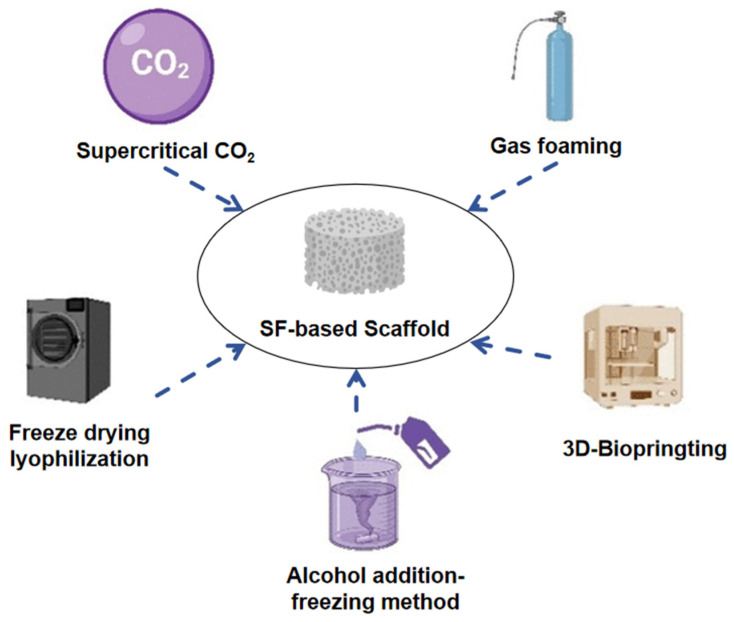
Preparation methods of SF-based porous scaffolds [97].

**Figure 7 materials-17-03924-f007:**
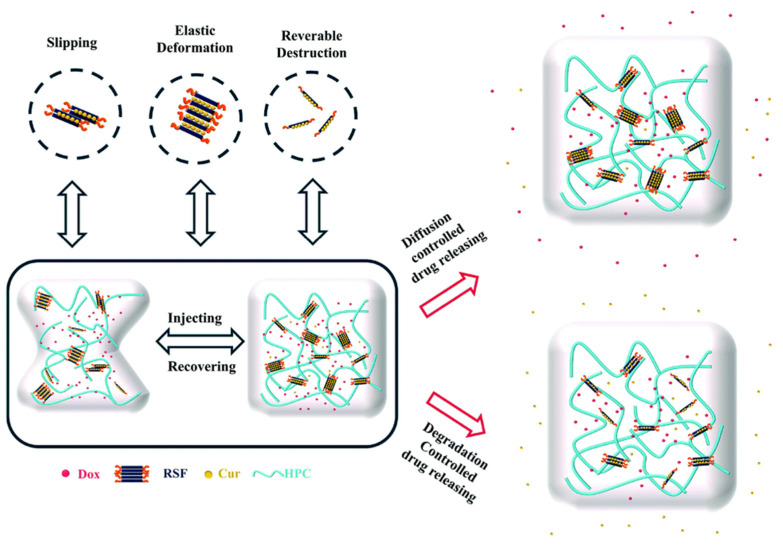
Mechanism of thiol-oxidation and drug release curve schematic in dual-loaded RSF/HPC hydrogel [112].

## Data Availability

No new data were created or analyzed in this study.

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
