# Peer review of "The Application of Regenerated Silk Fibroin in Tissue Repair"

_materials, 2024, doi:10.3390/ma17163924_

Round 1

Reviewer 1 Report

Comments and Suggestions for Authors

This review focuses on silk fibroin-based soft materials research, summarizes and introduces their preparation methods and basic applications, and their regenerative effects as drug delivery carriers on various aspects of tissue engineering such as bone, blood vessels, nerves, and skin in recent years. The current version of the paper requires significant modifications before it can be considered for publication in this journal. The following are the main observations:

1.     Section 2 could be titled “Stages involved in the wound repair process” since it mainly discusses that topic.

2.     The description in Figure 1 (page 3, lines 118-121) does not correspond to the figure inserted in the document. Please correct the description or insert the figure that corresponds to section 2.

3.     Including a representative diagram of the four stages involved in the wound repair process would be very didactic, providing the readers with a more informed and educated understanding of the manuscript.

4.     The authors indicated “The heavy chain, the most important component of silk fibroin…”. Please explain the particular characteristics that this polypeptide chain has to be considered “the most important component”. Include a reference for support.

Comments on the Quality of English Language

The document has several grammar and writing errors; please review it carefully before submitting it again.

Reviewer 2 Report

Comments and Suggestions for Authors

The manuscript of Li et al. describes the application of regenerated silk fibroin in tissue repair. Several papers discuss related applications of silk fibroin, and it would be interesting to highlight this review's focus (it was unclear to me). A table could provide a comparison of the performances of devices in the state-of-the-art.

Furthermore, the Abstract is composed of short (and repeated) sentences. The corresponding conclusion must be revised entirely. Section 2 is out of context.

Based on these comments, I believe a profound revision of the manuscript is a necessary and crucial step for the improvement of your work before a new evaluation.

Comments on the Quality of English Language

Revision in language is required.

Reviewer 3 Report

Comments and Suggestions for Authors

The scientific article “The Application of Regenerated Silk Fibroin in Tissue Repair” aimed to review soft materials based on silk fibroin, summarize and present their preparation methods and basic applications. It can be considered that:

1)      Rewrite sentences that are full copies of the articles used for the review. The similarity percentage for iThenticate is 35%, extremely high for a scientific manuscript.

2)      The abstract must be improved. It does not include the main findings of the review, their importance for the international scientific community and perspectives of the analyzed materials.

3)      Increase the number of keywords to facilitate searches in literature databases.

4)      In lines 36 and 37 (….By 2024, the global wound care market is expected to have a compound annual growth rate of 4.6%, increasing from $19.8 billion in 2019 to $24.8 billion...), insert the reference.

5)      In lines 50-54, insert the reference.

6)      In "1.3 Current Soft Materials Used in Tissue Repair" and "1.4 Advantages of Silk Fibroin Soft Materials in Tissue Repair" there is no reference. I request greater attention to the text of the entire manuscript so that references are correctly inserted.

7)      Were the figures created by the authors themselves?

8)      All abbreviations used must be preceded by their meaning. Please review the text. For example, CCN1 on line 170.

9)      Calls in the text of figures must be made before inserting them into the manuscript. Correct in figure 4.

10)  Insert possible limitations of the study.

Comments on the Quality of English Language

Minor editing

Reviewer 4 Report

Comments and Suggestions for Authors

The authors reported a review entitled “The Application of Regenerated Silk Fibroin in Tissue Repair”.

The review is well written and organized. It displays a very comprehensive background on the Silk Fibroin-based materials and the methods known in literature for preparing these soft materials.

The figures are well designed, and their quality is very high.

The discussed topic is of actual interest for the scientific community.

The presented materials and the techniques for realizing these materials could be helpful for scientists who working in wound healing and tissue engineering fields.

However, in the following, there are just a few suggestions for improving the manuscript before the publication:

In sub-sections “1.3 Current Soft Materials Used in Tissue Repair”, and “1.4 Advantages of Silk Fibroin Soft Materials in Tissue Repair”, it should be better to report some references related to the mentioned materials.

The same suggestion for the following sub-sections: “4.1 Preparation of Silk Fibroin Films”, when referring to the sentences reported at lines 361-364; sub-section “4.3 Porous Scaffolds”; sub-sub-section “4.3.3 Gas Foaming”; sub-sub-section “4.3.4 Supercritical CO2 Processing”.

In sub-section “2.2 Inflammatory Phase”, the following abbreviations needs to be explained: “Rac1” (line 169), “CCN1” (line 170), as well as the following along the review: “BMSC” (line 611), “JNK” “MEK”PI3K” (line 824), “Rb1/TGF-β1” (line 832).

In sub-sub section “4.3.5 3D Bioprinting”, please, revise the English form of the expression “polymer material” reported at line 561; it should be reported as “polymeric material”.

Please, check those references 101, 102, which are missing along the review.

In the “References”, please, check reference 89. 

Comments on the Quality of English Language

In sub-sub section “4.3.5 3D Bioprinting”, please, revise the English form of the expression “polymer material” reported at line 561; it should be reported as “polymeric material”.

Round 2

Reviewer 1 Report

Comments and Suggestions for Authors

The current version of the document is much improved. The authors have addressed each of the comments made. I consider that the article should be approved for publication.

Comments on the Quality of English Language

The document contains some grammar and writing errors.

Reviewer 2 Report

Comments and Suggestions for Authors

In view of the modifications provided by the authors, the manuscript can be accepted as is.

Reviewer 3 Report

Comments and Suggestions for Authors

No comments

Comments on the Quality of English Language

Minor editing